# Detection and Inhibition of *Clostridium perfringens* by Cocktail of Star Anise and Thymus Extracts in Chicken Meat Products

**DOI:** 10.3390/pathogens14060552

**Published:** 2025-06-01

**Authors:** Gamal M. Hamad, Shenoda Gaber Monir Henry, Gamal E. A. El-Rokh, Nadia H. A. Ramadan, Hany S. Abdel Raoof, Ahmed M. Sulaiman, Ahmed M. El-Mesallamy, Samy E. Elshaer, Sara M. Gaber, Ibrahim M. Rabah, Ahmed R. Mahmoud, Mahmoud S. A. Salama, Taha Mehany, Hesham E. A. Abdelfttah

**Affiliations:** 1Food Technology Department, Arid Lands Cultivation Research Institute, City of Scientific Research and Technological Applications, New Borg El Arab 21934, Egypt; sara.mgaber@gmail.com (S.M.G.); tmehany@srtacity.sci.eg (T.M.); 2Department of Food Science and Technology, Faculty of Agriculture, Assiut University, Assiut 71511, Egypt; shenoda_henrry@agr.aun.edu.eg (S.G.M.H.); nadiahelal@aun.edu.eg (N.H.A.R.); 3Department of Food Science and Technology, Faculty of Agriculture, Al-Azhar University, Assiut 71524, Egypt; dr.gamal83@gmail.com (G.E.A.E.-R.); drhanysalama12@gmail.com (H.S.A.R.); dr.ahmedmansour@azhar.edu.eg (A.M.S.); dr.ahmedabdo@azhar.edu.eg (A.M.E.-M.); d.ahmed_rashad@yahoo.com (A.R.M.); dr.salama@azhar.edu.eg (M.S.A.S.); heshamabdel-mobdy4919@azhar.eud.eg (H.E.A.A.); 4Department of Environmental Studies, Institute of Graduate Studies and Research, Alexandria University, Alexandria 21568, Egypt; samyelshaer2@gmail.com; 5Department of Animal Hygiene and Zoonoses, Faculty of Veterinary Medicine, Matrouh University, Matrouh 51744, Egypt; ibrahim.rabah@mau.edu.eg

**Keywords:** *Clostridium perfringens*, chicken meat, detection, inhibition, star anise, thymus extracts, chicken meat products, food preservatives

## Abstract

Background: The presence of *C. perfringens* in chicken meat products leads to significant economic losses for the industry and is associated with human food poisoning. It inspires creative answers for novel substances and techniques, such natural antibacterials, for improved prospects in the future. The objective of this study was to examine the antibacterial properties of thymus and star anise extracts for the suppression of *Clostridium perfringens* in products made from chicken meat. Methods: Thymus and star anise extracts were phytochemically analyzed using the Folin–Ciocalteu spectrophotometric method, High-Performance Liquid Chromatography (HPLC) to determine the phenolic compounds, DPPH to determine the antioxidant activity, and the agar disk diffusion assay to determine the antibacterial effect of the star anise. Following treatment with the mixture extract, an experimental application was conducted on chicken burgers. *C. perfringens* was also found in poultry samples. Lastly, the sensory evaluation of the chicken burger was detected. Results: The findings showed that *Clostridium perfringens* was present in a large number of chicken meat products, with a significant concentration in chicken thighs (84%), as well as in the mixture extract has the largest concentration of phytochemical components (TPC:123.88 ± 7.42 mg GAE/g, TFC: 69.04 ± 6.37 mg quercetin/g) with strong antibacterial action (the minimum inhibitory concentration (MIC) of the mixed extract was 3.12 mg/mL, with an inhibition zone of 13.06 ± 0.91 mm). Also, the addition of mix. extract to chicken burgers extends their shelf life, particularly when the mix content is high. Additionally, the mix. extract shows non-significant effect (*p* < 0.05) on the chicken burger’s sensory qualities. Conclusions: Finally, we can make use of the mix. extract of thymus and star anise as a natural preservative in chicken meat products especially the chicken burger.

## 1. Introduction

Chicken meat is a widely consumed protein source in Egypt, due to its affordability, nutritional benefits, and appealing flavor. Its high consumption is supported by the availability of various forms, including raw, pre-packaged, and ready-to-eat products, which are known for their consistent quality and ease of preparation [1]. Any lapse in sanitary protocols at slaughterhouses or processing facilities can lead to microbial contamination, posing significant health risks to consumers [2]. One such contaminant is *Clostridium perfringens*, a Gram-positive, non-motile, rod-shaped, and spore-forming bacterium commonly found in diverse environments and the normal gut flora of humans and animals [3,4]. *C. perfringens* can spread through various pathways, including contaminated food and water, animal contact, and person-to-person transmission [5]. Cross-contamination during food processing and poor hygiene practices are major contributors to its prevalence. The bacterium’s ability to form spores allows it to survive harsh conditions, such as exposure to oxygen and food manufacturing processes, making it a persistent food safety challenge [6,7]. Once ingested, *C. perfringens* can cause foodborne illnesses, including food poisoning, gas gangrene, and infectious diarrhea. Its spores can withstand stomach acidity, allowing vegetative cells to reach the small intestine, where they sporulate and release enterotoxins, leading to gastrointestinal symptoms [8].

One potential strategy to mitigate bacterial contamination and extend the shelf life of meat products is the use of antioxidants, which help slow lipid oxidation by neutralizing free radicals [9,10]. In addition, antioxidants are widely incorporated into both fresh and cooked meats to preserve quality and safety [10,11]. Natural antioxidants derived from fruits, vegetables, seeds, herbs, cereals, and spices have been extensively studied for their efficacy in food preservation [12,13].

Aromatic plants (e.g., thyme and star anise) are well appreciated not only for flavor, knowledge which is steeped in tradition, but also for their multifaceted bioactive properties. Unlike synthetic agents, these natural extracts possess antioxidant and antimicrobial actions, which can be attributed in a large part to the presence of various bioactive components such as alkaloids, saponins, tannins, terpenoids, polyphenols, flavonoids and essential oils [14,15,16]. Star anise (*Pimpinella anisum*), traditionally used as a culinary ingredient, is recognized for its rich composition of bioactive compounds with potent antioxidant properties [17] due to the presence of primary active components including anethole (90%), limonene (3.75%), and 4-allylanisole (1.5%), which contribute to its strong antioxidant and metal-chelating capabilities [18]. Similarly, thyme (*Thymus* spp.), a genus within the Lamiaceae family, is widely cultivated in Europe, the Mediterranean region, and parts of Asia, South America, and Australia. Numerous studies have demonstrated its antibacterial properties, making it a promising candidate for food preservation [19,20]. The antimicrobial effects of various Thymus species have been extensively analyzed, highlighting their potential as natural preservatives against Gram-negative (*E. coli*, *P. aeruginosa*) and Gram-positive (*S. aureus*, *B. subtilis*) bacteria, which are known to cause food spoilage and poisoning [21,22]. Beyond food preservation, bioactive compounds from plants have shown potential applications in medicine, including antibacterial and anticancer therapies [23,24]. These compounds can function as standalone antimicrobial agents, although the development of multi-target therapeutic formulations remains an area of ongoing research [25].

Despite numerous studies on each extract individually, the rationale behind combining *Thymus serpyllum* and star anise lies in their complementary phytochemical profiles. Star anise provides potent phenolic acids and flavonoids, while thyme contributes strong antimicrobial terpenoids and rosmarinic acid. This combination may result in synergistic effects, enhancing overall antibacterial and antioxidant activity, especially against spore-forming pathogens like *C. perfringens*. Furthermore, the blend could offer broader-spectrum protection while minimizing the concentration needed for each extract, which may help preserve sensory quality in food products.

This study aims to investigate (1) the prevalence of *Clostridium perfringens* in various chicken meat products available on the market; (2) the antimicrobial and antioxidant properties of star anise and thyme extracts—both individually and in combination—against *C. perfringens*, including their application in formulated chicken burgers; and (3) to determine the sensory impact of incorporating the mixed extract into chicken products, with a focus on acceptability. Within this framework, we examine the integrated use of star anise and thyme extracts as natural preservatives rather than positioning antioxidants as standalone antimicrobial agents.

## 2. Materials and Methods

### 2.1. Materials and Collection of Samples

The Star Anise and Thymus were purchased from local market and were ground into powder for preparation of extracts. For bacteriological detection and sampling of *C. perfringens*, 300 randomly selected chicken meat products were gathered from various retail grocery stores and supermarkets in the Alexandria Governorate, Egypt. A total of 50 from each of the following were gathered: raw chicken breast, raw chicken thigh, and partially cooked chicken products (nuggets, panée, frankfurter, burger). For sensorial study, minced breast chicken purchased from a local poultry shop in Alexandria, Egypt, was used for chicken burger preparation. A reference strain of *Clostridium perfringens* EMCC1574 anaerobically grown on Reinforced Clostridial Medium (RCM) agar (Oxoid, UK) was provided by Microbiological Resources Center (MERCIN), Faculty of Agriculture, Ain Shams University, Egypt. EMCC1574 was kept in RCM at −80 °C with 20% glycerol according to [26] until further usage.

### 2.2. Star Anise and Thymus Extracts Preparation

Ten grams of the ground powder were dissolved in 100 mL of 70% ethanol (*v*/*v*), forming the total mixture volume. The solution was shaken at 100 rpm and incubated at room temperature for 48 h. After centrifugation at 2.47× *g* for 30 min and filtration, the extracts were lyophilized at −50 °C using a Telstar Freeze-dryer Model 50 (France). The lyophilized extract was reconstituted in distilled water to known concentrations (mg/mL); the mixture extract was prepared by adding equal concentrations of thymus and star anise (1:1) and stored at refrigeration temperature (storage condition: 4 °C) until further analysis as described by [5].

### 2.3. Phytochemical Analysis of Extracts

**Total phenolic content (TPC).** The total phenolic content of Thymus and star anise extracts and the mixture of them (1:1) was determined and replicated three times using the Folin–Ciocalteu spectrophotometric method [27]. Two milliliters (1%) of the extracts were mixed with 0.1 mL of Folin–Ciocalteu reagent and allowed to stand for 15 min. Then, 3 mL of 2% sodium carbonate (Na_2_CO_3_) was added. After 30 min at room temperature, absorbance was measured at 760 nm using a T80 UV/VIS spectrophotometer (PG Instrument Ltd., Lutterworth, UK). A standard calibration curve of gallic acid was used to quantify total phenolic content, expressed as milligrams of gallic acid equivalent (GAE) per gram of sample. Total flavonoid content. The total flavonoid content of the extracts was determined using a colorimetric method as described by [28]. Absorbance was measured at 510 nm using a T80 UV/VIS spectrophotometer (PG Instrument Ltd., UK). Results were expressed as milligrams of quercetin equivalent per gram of sample. Phenolic and flavonoids compounds. Phenolic compounds were analyzed using an Agilent 1260 Infinity HPLC system (Agilent, Santa Clara, CA, USA), equipped with a quaternary pump and a Zorbax Eclipse Plus C18 column (100 mm × 4.6 mm, Agilent Technologies, Santa Clara, CA, USA). The column was maintained at 25 °C. Separation was performed using a ternary linear gradient elution system consisting of (A) methanol, (B) acetonitrile, and (C) HPLC-grade water containing 0.2% phosphoric acid (H_3_PO_4_, *v*/*v*). The injection volume was 20 µL, and detection was carried out using a variable wavelength detector (VWD) set at 284 nm. The method followed the Agilent Application Note (Publication number 5991-3801 EN, 2016) [29].

**Antioxidant activity (DPPH Assay).** The antioxidant activity of Thymus and the star anise extracts was evaluated using the DPPH radical scavenging assay. This method assesses the ability of the extracts to neutralize the DPPH radical [30,31]. Extracts were tested at various concentrations (5, 10, 20, 30, 40, 50, 60, 70, 80, and 100 mg/mL), which were prepared from lyophilized powder obtained by extracting 10 g of each dried plant material (Star Anise and Thymus) in 100 mL of 70% ethanol. Ascorbic acid at equivalent concentrations was used as a standard reference. The absorbance was measured at 517 nm using a T80 UV/VIS spectrophotometer (PG Instrument Ltd., Wibtoft, UK). DPPH assay was used in this study as a rapid and widely accepted method to evaluate the free radical scavenging capacity of the plant extracts. This is not intended to suggest that antioxidant capacity alone explains the antimicrobial effects observed but rather to complement the phytochemical analysis and help characterize the bioactive potential of the extracts.

### 2.4. Anti-Clostridial Activity of Extracts

The agar disk diffusion assay was used to evaluate the antibacterial activity of Thymus and star anise extracts alone or in mixture (1:1) replicated three times. against the reference strain *C. perfringens* EMCC1574 [32]. An overnight culture of *C. perfringens* was diluted to a concentration of 10^6^ CFU/mL and supplemented with Reinforced Clostridial Medium (RCM) broth (Oxoid, UK). The culture was incubated at 37 °C for 48 h to achieve semi-confluent growth. Subsequently, the bacterial suspension was evenly spread onto RCM agar plates using sterile cotton swabs. After allowing the plates to dry, 20 µL of Thymus and star anise extracts were individually applied to sterile filter paper disks. The plates were then maintained at 4 °C for 30 min to facilitate proper diffusion before being incubated at 37 °C for 24 h under anaerobic conditions. Following incubation, the inhibition zones were measured in millimeters to assess the anti-clostridial activity of the extracts. For comparative analysis, antibiotic disks containing Oxytetracycline (OX), Erythromycin (E), Cefadroxil (CFR), Cefazolin (CZ), Doxycycline (DO), Metronidazole (MT), Roxithromycin (RoX), Clindamycin (DA) (2 µg/disc), Tetracycline (TE) (30 µg/disc), Amoxicillin (AX) (25 µg/disc), Cefoxitin (CX) (30 µg/disc), Chloramphenicol (C) (30 µg/disc), and Cefotaxime (CTX) (30 µg/disc) were included as controls [33].

### 2.5. Identification of Minimum Inhibitory Concentrations (MICs)

Only the mixture of thymus and star anise extract (1:1) was tested for MIC against *C. perfringens* using a serial dilution method replicated three times. *C. perfringens* suspensions were prepared in sterile saline from actively growing cultures and adjusted to a density of 10^6^ CFU/mL. A sterile cotton swab was used to evenly inoculate RCM agar plates (Oxoid, UK) with the bacterial suspension. The plates were left to dry at room temperature for 15 min before the application of test disks. Eight sterile filter paper disks (6 mm in diameter) were placed on the agar surface. Each disk was impregnated with 10 µL of serially diluted mixed extract prepared in Milli-Q water. The inoculated plates were incubated at 37 °C for 24 h under anaerobic conditions. The MIC values were recorded as the lowest concentration of the extract that produced a visible inhibition zone. Each test was performed in triplicate for accuracy [34].

### 2.6. Detection of C. perfringens in Chicken Samples

Chicken samples were chopped into small pieces using sterile scissors and were serially diluted with autoclaved 0.1% peptone water (1:9 *w*/*v*), then shaken for 2 min at 10,000× *g* (Wise-Shake-Wisd-SHR-2D- Republic of Korea) to homogenize them. The homogenates were heated at 80 °C for 15 min in a water bath, then 1 mL of each homogenate was deeply inoculated with 9 mL of Fluid Thioglycolate Medium (Merck KGaA, Darmstadt, Germany) and incubated at 37 °C for 24 h. Egg yolk tellurite emulsion (Oxoid Ltd., Basingstoke, UK) was added to Reinforced Clostridial Medium agar after a loopful (10 mL) of thioglycolate inoculum had been streaked onto it in a sterile Petri dish. Following observation of the plates’ growth, the inoculated agar plates were incubated at 37 °C for 48 h in a CO_2_ incubator (New Brunswick™ Galaxy^®^ 170 R CO_2_ Incubator Series, Eppendorf, Hamburg, Germany). Black colonies exhibiting characteristics of *C. perfringens* were counted in positive samples, as described by [32].

### 2.7. Experimental Application on Chicken Burger After Treatment by Mixture Extract

The potential development of *Clostridium perfringens* in a chicken burger was tested, whether they were originally infected or not. Table 1 lists the ingredients used to manufacture the chicken burger, which included 88% minced breast chicken, 5% whole egg, 1% salt, 1% black pepper, and 5% breadcrumbs as the base, and the mix extract was applied to the chicken burger at several amounts (0.5, 1%, and 2%). The negative control was the chicken burger as is without added *Clostridium perfringens* or mix extract and positive control was a samples of *Clostridium perfringens* (10^6^ CFU/mL) and mix extract were used as positive controls. The samples were aerobically packed in bags from polypropylene trays that were made for single-use food packaging and kept at 4 °C after being individually prepared in a bowl chopper to create chicken burgers. Raw chicken burger samples were utilized for analysis microbiological count to *Clostridium perfringens* at 0, 1, 2, 3. 4, 5, 6, and 7 days of keeping things in a refrigerator (4 °C). Additionally, the sensory characteristics of cooked chicken burger samples (treatment and negative control) were ascertained. Every experiment was run in triplicate [34,35].

### 2.8. Sensory Evaluation of Chicken Burger

At the Food Technology Department of the City of Scientific Research and Technological Applications in New Borg El Arab, Egypt, a panel of ten judges with experience in the qualities of chicken products conducted a sensory evaluation. A descriptive 9-point scale was used to assess various sensory qualities, such as texture, flavor, taste, color, and overall acceptability [36], where 1 indicates a strong dislike and 9 indicates a strong liking. Before being served to the panelists for evaluation, the chicken patties were warmed in a microwave oven for 50 s and cooked for 5 min on a grill until the center of the chicken reached 80 °C. In between samples, mouthwash was administered with water. Three treatment groups with increasing doses of mix extract (0.5%, 1%, and 2%), as well as a control group that received no treatment (chicken burgers without mix extract), were used for the evaluation. The information about sensory attributes, including standard deviations, were analyzed and recorded according to [35,37].

### 2.9. Statistical Analyses

All experiments were conducted in triplicate (*n* = 3). Data are presented as mean ± standard deviation (SD). Statistical analysis was performed using one-way analysis of variance (ANOVA), followed by Duncan’s multiple range test for post hoc comparisons. Differences were considered statistically significant at *p* < 0.05 or *p* < 0.01, as indicated. All statistical analyses were carried out using IBM SPSS Statistics version 23.

## 3. Results and Discussion

### 3.1. Incidence of C. perfringens in the Examined Chicken Meat Product Samples

The analysis of raw and half-cooked chicken samples showed high *C. perfringens* prevalence, with chicken thighs (84%) and breasts (72%) most affected among raw samples (Figure 1). In half-cooked products, chicken panée (76%) had the highest incidence, followed by nuggets (70%), burgers (68%), and frankfurters (66%). Therefore, chicken thighs are considered the most contaminated chicken species with *C. perfringens* compared to all the species under study.

These findings align with [4] who detected *C. perfringens* in 94% of wing samples, 80% of leg quarters, 66% of drumsticks, and 66% of breasts. Ref. [38] also reported significant contamination in chicken products, while [39] found it in 69.6% of raw chicken livers. The high prevalence in raw and partially cooked chicken may impact food safety and quality [40], with contamination linked to poultry intestinal flora, slaughter-related cross-contamination, and poor hygiene [41].

### 3.2. The Use of Star Anise and Thymus Extracts as Antimicrobial Agents

#### 3.2.1. Phytochemical Composition of Star Anise and Thymus Extracts and Their Mixture

Phytochemical analysis revealed that the star anise extract had significantly higher total phenolic content (TPC) and total flavonoid content (TFC) compared to thymus extract (Table 2). The combination of both extracts resulted in the highest TPC (123.88 ± 7.42 mg GAE/g) and TFC (69.04 ± 6.37 mg quercetin/g). Analysis of the polyphenolic composition (Table 3) showed that star anise extract was particularly rich in gallic acid (35.35 µg/g), catechin (85.53 µg/g), rutin (67.91 µg/g), and coumaric acid (74.47 µg/g). In contrast, thymus extract contained significantly higher concentrations of rosmarinic acid (150.44 µg/g), salvianolic acid (27.26 µg/g), and caffeic acid (15.52 µg/g). These findings align with previous studies on Thymus species [22] and star anise [42], which confirm their rich phenolic profiles. These results are consistent with previous studies indicating that Thymus plants contain a diverse range of polyphenolic compounds, particularly flavonoids and phenolic acids, known for their strong antioxidant activity [43]. Similar TPC and TFC values were reported for different solvent extracts of Thymus species [44,45]. In the case of star anise, the presence of quercetin, gallic acid, rutin, caffeic acid, and chlorogenic acid, supporting its known bioactivity were also reported [42,46]. Although thymus contains bioactive compounds such as terpenoids and tannins, its antimicrobial activity is mainly attributed to flavonoids. These compounds interact with bacterial cell walls and extracellular proteins, while lipophilic flavonoids may also disrupt bacterial membranes [15,47]. Similarly, star anise extract has demonstrated antimicrobial properties, which may be attributed to its high content of phenolic acids and flavonoids, compounds known to exhibit antibacterial effects by disrupting microbial cell structures and inhibiting essential enzymes.

#### 3.2.2. Antioxidant Activity and DPPH Radical Scavenging Capacity of Extracts

The DPPH radical scavenging assay was used to evaluate the antioxidant activity of Thymus, star anise, and their mixture, with ascorbic acid serving as the reference standard (Figure 2). The mixture extract exhibited significantly higher antioxidant activity (IC50: 34.87 ± 0.71 µg/mL) compared to star anise (IC50: 47.91 ± 1.91 µg/mL) and Thymus (IC50: 62.41 ± 1.14 µg/mL), though ascorbic acid remained the most potent (IC50: 18.67 ± 0.71 µg/mL). The results indicated that the extracts of a mixture of thyme and star anise, they have proven their antibacterial activity due to their high content of phenolic and flavonoid compounds. Therefore, they can be used to preserve food instead of chemical additives that have side effects on humans.

These findings are consistent with [45], who demonstrated the antioxidant potential of different extracts using DPPH, reporting an IC50 of 0.416 mg/mL for the ethanol fraction Reference. [48] found that star anise extract exhibited significant antioxidant activity (IC50 = 3.46%), suggesting that its polyphenolic composition, particularly flavonoids and phenolic acids, plays a crucial role in its free radical scavenging capacity. The antioxidant properties of star anise essential oil (SAO) have been well documented, further confirming its free radical scavenging and lipid peroxidation inhibition effects, along with its antifungal activity against various fungal species. The main bioactive compound in star anise essential oil, trans-anethole, is believed to contribute to these effects due to its double bonds, which enhance radical scavenging activity [49]. Additionally, synergistic interactions between multiple components in star anise essential oil may further amplify its antioxidant properties. Similarly, studies on Thymus species indicate a strong DPPH radical scavenging capacity, with flavonoid-rich extracts exhibiting higher activity than hydroalcoholic extracts [50]. For instance, *Thymus kotschyanus* (IC50: 47.22 µg/mL), *Thymus daenensis* (IC50: 48.68 µg/mL), and *Thymus pubescens* (IC50: 31.47 µg/mL) have demonstrated notable antioxidant effects.

#### 3.2.3. Influence of Extracts on *C. perfringens* Activity

The mixture of star anise and thymus extracts exhibited the strongest antibacterial activity, with an inhibition zone of 31.07 ± 2.01 mm, followed by star anise extract (26.07 ± 1.06 mm) and thymus extract (23.13 ± 1.04 mm). This enhanced effect is likely due to high levels of total phenolics and flavonoids. The minimum inhibitory concentration (MIC) of the mixed extract was 3.12 mg/mL, with an inhibition zone of 13.06 ± 0.91 mm (Table 4, Figure 3). The most effective antibiotics were Ox, DO, RoX, TE, and DA. Compared to these drugs, the extracts demonstrated strong antibacterial action. These findings align with literature [51], where star anise’s bioactive components with strong antibacterial, antioxidant, and anticancer properties, have been reported making it a promising ingredient for food and pharmaceutical applications. This is most probably due to their phenolic and flavonoid content [42]. Thymus contains flavonoids, terpenoids, tannins, and polyphenolic compounds, which contribute to its antimicrobial, anticancer, and antioxidant properties [52]. The observed variation in inhibition zones may be due to differences in diffusion rates of these bioactive components.

### 3.3. Application of Extracts in Chicken Burger

#### 3.3.1. Influence on Development of *C. perfringens* During Storage

During the preparation and processing of chicken burgers, different concentrations of mix extract (1%, 1.5%, and 2%) were tested to evaluate their effect on shelf-life improvement. The results presented in Figure 4 indicate that *C. perfringens* was completely inhibited in all treated samples, regardless of the extract concentration. This finding suggests that the applied extract enhanced microbiological quality in a concentration-dependent manner, with the highest inhibition observed at 2%. These results align with the findings of AA. Aly., 2019 [53] which concluded that pomegranate peel extract could serve as a natural food additive to improve the microbial safety of chicken products due to its antioxidant and antibacterial properties. Star anise extract was also reported as potential replacer of butylated hydroxytoluene in stabilizing cooked pork patties during refrigerated storage while also exhibiting bacteriostatic effects [54]. While incorporating essential oils from thyme, rosemary, sage, marjoram, and black seeds into chicken burgers were reported to extend their shelf life up to 180 days under frozen storage at −18 °C [55].

#### 3.3.2. Sensory Evaluation of Chicken Burger Fortified with Mix Extract

Compared to the control, the sensory characteristics of chicken burgers treated with the extract mixture showed no significant differences in color, odor, texture, taste, appearance, or overall acceptability across all tested concentrations, Figure 5. The addition of rosemary, basil, and mint leaf extracts to chicken burgers were previously reported to not significantly influence the burger sensory qualities [56]. The incorporation of other leaf source such as Moringa oleifera as a natural food preservative were also reported to not significantly impact chicken patty products [35].

## 4. Conclusions

The study highlights the potent antibacterial and antioxidant properties of star anise and thymus extracts, both individually and in combination. The extract mixture demonstrated the highest total phenolic and flavonoid content, correlating with enhanced antibacterial activity against *C. perfringens*, surpassing the effects of synthetic antibiotics. Furthermore, the extract mixture exhibited strong radical scavenging capacity, reinforcing its potential as a natural antioxidant. The high prevalence of *C. perfringens* in raw and partially cooked chicken meat products underscores the need for effective antimicrobial strategies. The blend of star anise and thymus extracts significantly reduced *C. perfringens* growth in chicken meat, offering a promising natural preservative alternative. Although its impact on sensory attributes was minimal, its effectiveness in enhancing food safety suggests potential applications in meat preservation and functional food development.

## Figures and Tables

**Figure 1 pathogens-14-00552-f001:**
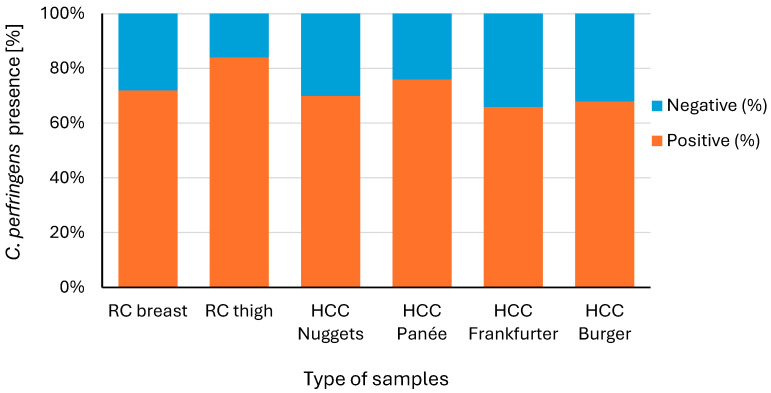
Percentage of *C. perfringens* colonies present in the examined chicken meat products out of the total number per samples (*n* = 50). RC: raw chicken meat, HCC: Half-cooked chicken products.

**Figure 2 pathogens-14-00552-f002:**
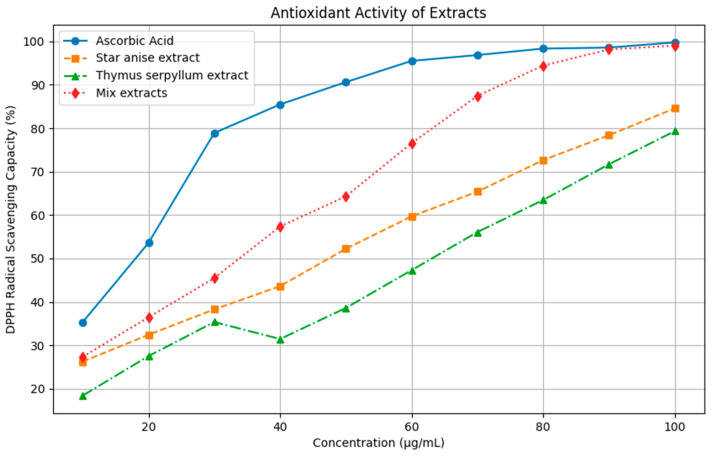
DPPH radical scavenging capacity of Star anise extract, *Thymus serpyllum* extract and both extracts mixture (1:1) at different extracts concentration (µg/mL). Data presents means of triplicates with standard deviations.

**Figure 3 pathogens-14-00552-f003:**
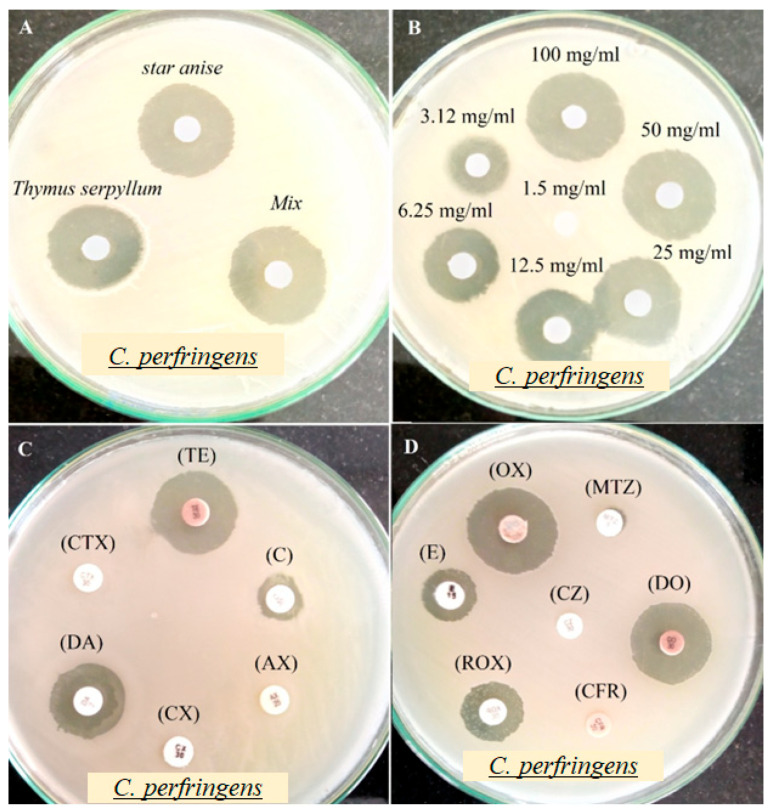
(**A**) Antibacterial activity of Star anise and *Thymus serpyllum* extracts, (**B**) minimum inhibitory concentrations (MICs) of *Mix Extract*, (**C**,**D**) antibiotic, against *Clostridium perfringens* using agar disk diffusion assay.

**Figure 4 pathogens-14-00552-f004:**
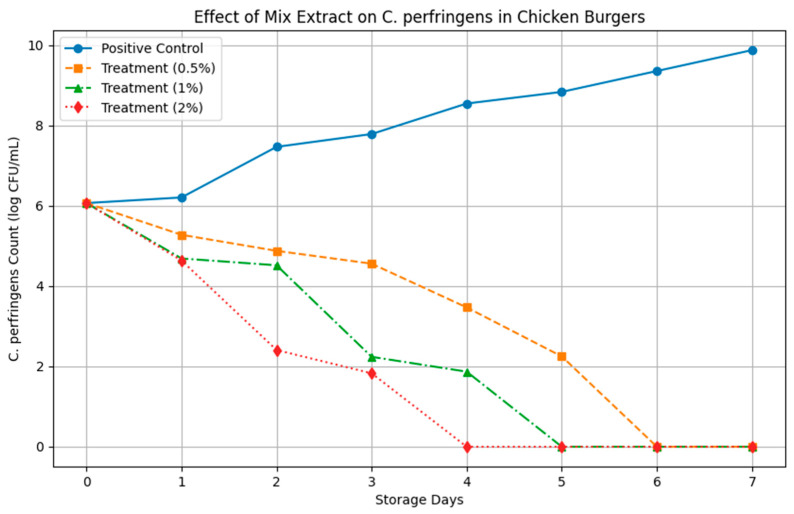
The impact of mix extract at different concentrations as an antibacterial effect against *Clostridium perfringens* (10^6^ cfu /mL) on shelf-life of chicken burger stored at 10 °C (mean ± SD). Control: untreated chicken burger, treatment (0.5%): mixture extract-treated chicken burger, treatment (1%): mixture extract-treated chicken burger, treatment (2%): mixture extract-treated chicken burger.

**Figure 5 pathogens-14-00552-f005:**
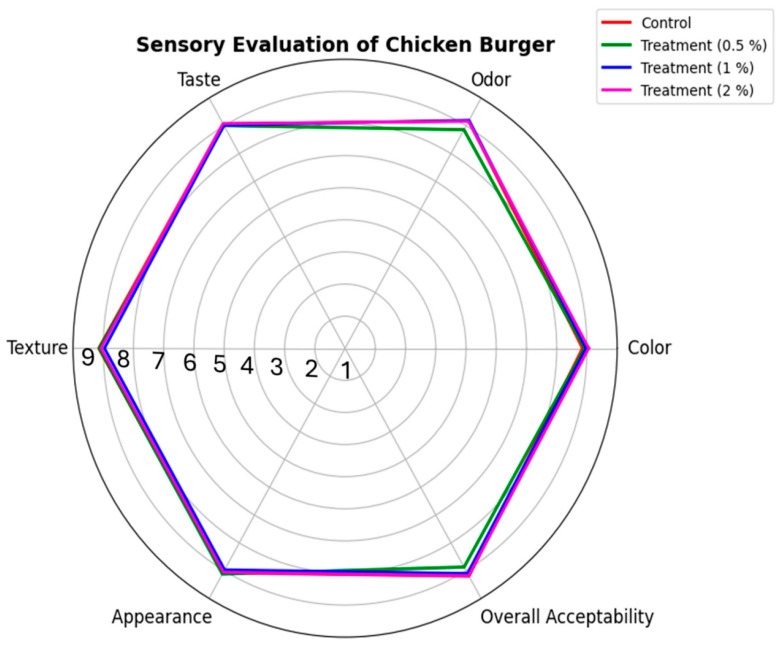
Sensory evaluation of chicken burger fortified with mix extract.

**Table 1 pathogens-14-00552-t001:** Recipe ingredients of chicken burgers with control and treatments.

Ingredients (%)	Control	Treatment		
		0.5% Mix Extract	1% Mix Extract	2% Mix Extract
Brest Chicken	88	88	88	88
Breadcrumbs	5.0	4.5	4.0	3.0
Whole egg	5.0	5.0	5.0	5.0
Salt	1.0	1.0	1.0	1.0
Black pepper	1.0	1.0	1.0	1.0
Mix extract	0.0	0.5	1.0	2.0

Treatments: control = chicken burgers (without Mix extract); 0.5% Mix extract = chicken burgers with 0.5% Mix extract; 1% Mix extract = chicken burgers with 1% Mix extract; 2% Mix extract = chicken burgers with 2% Mix extract.

**Table 2 pathogens-14-00552-t002:** Phytochemical composition of Total phenolic and Total flavonoids for Star anise, *Thymus serpyllum*, and Mix extracts.

Extracts	Total Phenolic Concentration (mg GAE/g)	Total Flavonoids Concentartaion (mg Quercetin/g)
*Star anise extract*	97.02 ± 6.63 ^b^	56.74 ± 5.37 ^b^
*Thymus serpyllum extract*	23.24 ± 0.84 ^c^	5.41 ± 1.04 ^c^
*Mix extract*	123.88 ± 7.42 ^a^	69.04 ± 6.37 ^a^

Different superiors on the same column show significant differences (*p* < 0.05).

**Table 3 pathogens-14-00552-t003:** Phenolic and flavonoids compounds of Star anise and *Thymus serpyllum* extract.

Phenolic and Flavonoid Compounds	Star Anise Extract [µg/g]	*Thymus serpyllum* [µg/g]
Gallic acid	35.35 ± 4.10 ^A^	2.02 ± 0.33 ^B^
Chlorogenic acid	8.74 ± 2.21 ^A^	0.00 ± 0.00 ^B^
Vanillic acid	8.84 ± 1.66 ^A^	0.00 ± 0.00 ^B^
Catechin	85.53 ± 5.30 ^A^	0.00 ± 0.00 ^B^
Cinnamic acid	10.84 ± 2.10 ^A^	7.91 ± 0.60 ^B^
Salicylic acid	3.71 ± 0.56 ^A^	0.00 ± 0.00 ^B^
Syringic acid	0.73 ± 0.42 ^A^	0.00 ± 0.00 ^B^
Ferulic acid	6.46 ± 2.06 ^B^	13.55 ± 2.15 ^A^
Rutin	67.91 ± 7.46 ^A^	0.00 ± 0.00 ^B^
Rosmarinic acid	0.00 ± 0.00 ^B^	150.44 ± 9.49 ^A^
Quercetin	5.36 ± 1.07 ^A^	0.00 ± 0.00 ^B^
Naringinin	0.00 ± 0.00 ^B^	2.90 ± 0.78 ^A^
Kaempferol	2.43 ± 0.68 ^A^	0.93 ± 0.37 ^B^
Apigenin	0.00 ± 0.00 ^B^	9.10 ± 0.20 ^A^
Caffeic acid	1.88 ± 0.55 ^B^	15.52 ± 1.06 ^A^
Ellagic acid	0.96 ± 0.56 ^A^	0.00 ± 0.00 ^B^
4-hydroxybenzoic acid	2.90 ± 0.45 ^A^	0.00 ± 0.00 ^B^
Coumaric acid	74.47 ± 9.19 ^A^	6.66 ± 2.15 ^B^
Sinapic	0.95 ± 0.39 ^A^	0.00 ± 0.00 ^B^
Luteolin	0.00 ± 0.00 ^B^	2.40 ± 0.50 ^A^
Benzoic acid	0.97± 0.49 ^A^	0.00 ± 0.00 ^B^
Myricetin	0.42 ± 1.11 ^A^	0.00 ± 0.00 ^B^
Epicatechin	0.00 ± 0.00 ^B^	0.50 ± 0.09 ^A^
Salvianolic acid	0.00 ± 0.00 ^B^	27.26 ± 1.56 ^A^

Mean with different capital superior letters in the same row present significant variations (*p* < 0.05).

**Table 4 pathogens-14-00552-t004:** Evaluation of antibacterial activity of Star anise and *Thymus serpyllum* extract compared to antibiotics and minimum inhibitory concentrations (MICs) for Mix extract against *Clostridium perfringens*.

Extracts/Antibiotics	Concentration	Inhibition Zone Diameter (mm) Mean ± SD
**Antibacterial activity of *star anise* and *Thymus serpyllum* extract**		
*Star anise extract*	100 mg/mL	26.07 ± 1.06 ^c^
*Thymus serpyllum extract*	100 mg/mL	23.13 ± 1.04 ^e^
*Mix extract*	100 mg/mL	31.07 ± 2.01 ^a^
Antibacterial activity of antibiotic		
*Oxytetracycline (Ox)*	30 µg/disc (0.030 mg)	22.10 ± 0.85 ^f^
*Erythromycin (E)*	50 µg/disc (0.050 mg)	12.13 ± 1.02 ^f^
*Cefadroxil (CFR)*	30 µg/disc (0.030 mg)	0.00 ± 0.00 ^l^
Cefazolin (CZ)	30 µg/disc (0.030 mg)	0.00 ± 0.00 ^l^
*Doxycycline (DO)*	30 µg/disc (0.030 mg)	20.13 ± 0.60 ^g^
*Metronidazole (MTZ)*	25 µg/disc (0.025 mg)	0.00 ± 0.00 ^l^
*Roxithromycin (RoX)*	30 µg/disc (0.030 mg)	17.06 ± 1.10 ^h^
*Tetracycline (TE)*	30 µg/disc (0.030 mg)	17.96 ± 0.95 ^h^
*Chloramphencol (C)*	30 µg/disc (0.010 mg)	11.00 ± 0.80 ^k^
*Amoxicillin (AX)*	25 µg/disc (0.025 mg)	0.00 ± 0.00 ^l^
*Clindamycin (DA)*	2 µg/disc (0.002 mg)	15.03 ± 0.94 ^i^
Cefoxitin (CX)	30 µg/disc (0.030 mg)	0.00 ± 0.00 ^l^
Cefotoxime (CTX)	30 µg/disc (0.030 mg)	0.00 ± 0.00 ^l^
**Minimum inhibitory concentrations (MICs) of Mix extract**		
*Mix Extract*	100 mg/mL	31.07 ± 2.01 ^a^
	50 mg/mL	27.16 ± 0.76 ^b^
	25 mg/mL	24.14 ± 0.91 ^d^
	12.5 mg/mL	20.30 ± 1.30 ^g^
	6.25 mg/mL	15.12 ± 0.96 ^i^
	3.12 mg/mL	13.06 ± 0.91 ^j^
	1.5 mg/mL	0.00 ± 0.00 ^l^

Significant differences are shown by different superiors on the same column (*p* < 0.05). Diameters include a 5 mm disc diameter, and MIC is the minimal inhibitory concentration in milligrams per milliliter.

## Data Availability

The data that support the findings of this study are available from the corresponding author upon reasonable request.

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
