# Peer review of "Detection and Inhibition of Clostridium perfringens by Cocktail of Star Anise and Thymus Extracts in Chicken Meat Products"

_pathogens, 2025, doi:10.3390/pathogens14060552_

Round 1
Reviewer 1 Report
Comments and Suggestions for Authors
Revise the abstract. There are several formatting issues. Unnecessary capitalization and several words are written with a hyphen. Include P-value in the abstract.
Introduction: What is the novelty of the study? This should be included. Several studies have been conducted on these antioxidants.
Indicate the number of replications in the stat section.
Line 182: Include details of packaging.
Check the number of the conclusion section. It is not 2.
Reviewer 2 Report
Comments and Suggestions for Authors
Abstract: Poorly written. The abstract contains several grammatical issues and unclear phrasing. The background and objective sentences are repetitive. No clear hypothesis is stated. Key quantitative results (e.g., MIC values, inhibition zones) should be summarized numerically. This section should be adjusted and rewritten.
Introduction: The introduction should be reviewed and worked on. Each paragraph should link with each other (missing linkage statements). There are too many grammatical errors and incomplete sentences, e.g., lines 85-86. Clearly state the rationale for using Thymus serpyllum + star anise combo.
Lines 65-69: Why the focus on antioxidants??
Methodology: Missing information such as the details on the mixture volume of the extracts.
lines 104-109: Provide details on missing key parameters (e.g., yield percentage, storage conditions).
Line 135: How many grams??? What particle size?
Line 138: a water bath to get rid of aerobic bacteria that didn't form spores?????
Lines 176-178: How was the extract applied to the nuggets??? Provide more details. Were the nuggets also infected with Clostridium?
Line 179: Details of the bags????
What quantity of each extract was mixed to determine the phytochemical analysis?
Discussion:
Lack of Critical Analysis: The authors mostly restate results rather than interpreting them in depth. There’s little effort to connect findings to underlying biological mechanisms (e.g., how phenolics disrupt bacterial membranes or influence spore germination). There’s no attempt to explain why the mixed extract outperformed individual ones beyond citing "high phenolic content."
The discussion needs more adjustments.
Reviewer 3 Report
Comments and Suggestions for Authors
The concept of the study is very interesting but the authors try to combine two many experiments in one.
There are three studies incorporated in one. The first is the market survey focusing on the detection of clostridium perfrigens on poultry products. The second one was the preparation of poultry products and the inhibition of clostridium perfrigens and the third one was the sensory evaluation.
Antioxidants are not used as antimocrobials. You use aromatic plants, herbs etc that have both antioxidant and antimicrobial function enabling the extension of shelf life. This is a point that should be included in the introduction.
Also it is surprising the fact that you measure antioxidant function but you did not estimate the extent of lipid oxidation.
My suggestion is to write again the manuscript. Do not use the data regarding DPPH.
In relation to the available data there should be a primary and a secondary aim of the study. The primary for example should be the antimicrobial effect whereas the secondary should be the market survey.
The outline of the manuscript should be
Incidence the Clostridium perfrigens in poultry products
The use of herbs and spices as antimicrobials.
Then provide your data regarding the characterization of the extracts, product preparation, etc.
Figure 9. Please add numbers in the axis.
In general, the entire manuscript should be written again,
Comments on the Quality of English Language
-
Round 2
Reviewer 1 Report
Comments and Suggestions for Authors
The authors have made changes in revised paper.
Reviewer 2 Report
Comments and Suggestions for Authors
Good
Reviewer 3 Report
Comments and Suggestions for Authors
-
Comments on the Quality of English Language
-